# Regionalization of Coarse Scale Soil Moisture Products Using Fine-Scale Vegetation Indices—Prospects and Case Study

**Mengyu Liang [1,*], Marion Pause [2] , Nikolas Prechtel [2] and Matthias Schramm [3]**

[1]  Department of Geographical Sciences, University of Maryland–College Park, College Park, MD 20740, USA
[2]  Faculty of Environmental Sciences, TU Dresden, 01062 Dresden, Germany;
    marion.pause@tu-dresden.de (M.P.); nikolas.prechtel@tu-dresden.de (N.P.)
[3]  Department of Geodesy and Geoinformation, TU Wien, 1040 Vienna, Austria;
    matthias.schramm@geo.tuwien.ac.at
*   Correspondence: mliang77@terpmail.umd.edu; Tel.: +1-301-405-4050

**Abstract:** Surface soil moisture (SSM) plays a critical role in many hydrological, biological and biogeochemical processes. It is relevant to farmers, scientists, and policymakers for making effective land management decisions. However, coarse spatial resolution and complex interactions of microwave radiation with surface roughness and vegetation structure present limitations within active remote sensing products to directly monitor soil moisture variations with sufficient detail. This paper discusses a strategy to use vegetation indices (VI) such as greenness, water stress, coverage, vigor, and growth dynamics, derived from Earth Observation (EO) data for an indirect characterization of SSM conditions. In this regional-scale study of a wetland environment, correlations between the coarse Advanced SCATterometer-Soil Water Index (ASCAT-SWI or SWI) product and statistical measurements of four vegetation indices from higher resolution Sentinel-2 data were analyzed. The results indicate that the mean value of Fraction of Absorbed Photosynthetically Active Radiation (FAPAR) correlates most strongly to the SWI and that the wet season vegetation traits show stronger linear relation to the SWI than during the dry season. The correlation between VIs and SWI was found to be independent of the underlying dominant vegetation classes which are not derived in real-time. Therefore, fine-scale vegetation information from optical satellite data convey the spatial heterogeneity missed by coarse synthetic aperture radar (SAR)-derived SSM products and is linked to the SSM condition underneath for regionalization purposes.

**Keywords:** surface soil moisture; regional scale; vegetation traits; multi-sensor approach; wetland; environmental monitoring

---

## 1. Introduction

Detecting surface soil moisture (SSM) is a key challenge in EO and of great interest in environmental monitoring at all spatial scales [1–5]. Advancements are available for SSM remote sensing and algorithms with a focus on large-scale (continental/global) applications [1–5]. Regional, intermediate-to-small-catchment scale soil-moisture monitoring can be performed without or independently of satellite remote sensing observations using techniques such as low-energy cosmic-ray neutrons and Proximal Gamma-Ray (PGR) spectroscopy [6,7]. However, it is of increasing importance for governmental agencies, scientists, and farmers to monitor SSM change in relations with climate and weather with fine spatial and temporal resolution remote sensing products [1–5], due to the fact that SSM tightly links to many hydrological, biological and biogeochemical processes at these finer scales. Thus, the mismatch between the availability and need of fine spatial resolution remote-sensing-based

SSM presents a clear gap for many environmental monitoring and applications on the one hand, and on the other hand, it highlights the importance of such information for local adaption efforts to mitigate the effects of climate change.

Due to its high significance in the Earth system, SSM has always solicited much attention in its measurement and monitoring using remote sensing techniques. SSM observations are retrieved from instruments sensing at microwave and optical/thermal infrared wavelength. Some key soil moisture products include the well-known ESA SMOS (Soil Moisture Ocean Salinity), SMAP (Soil Moisture Active Passive), and ASCAT-SWI [8–13]. However, most publicly available SSM datasets have coarse spatial resolutions ranging from 25–50 km [8–13]. A recent breakthrough is the 1 km Sentinel-1/ASCAT fusion product [13], which is currently available across Europe. Moreover, microwave radiation of soil water content is sensitive to surface roughness, as induced by a dynamic vegetation structure, for instance. As surface roughness increases or the vegetation canopy gets higher, the backscatter from differently polarized signals converges, whilst the noise level is getting higher [14,15]. Thus, in addition to the scale issue, traditional active remote-sensing-based SSM products cannot easily account for high variability in the terrain parameters and are affected by such noise when sensing landscapes with complex land cover and water patterns, for example, in the case of wetland areas.

Many efforts are currently taking place to explore the use of vegetation traits derived from passive remote sensing products for SSM monitoring. Alexandridis et al. (2016) adopted an integrated approach to derive evaporative fraction and saturated water content with the thermal infrared data from MODIS, in combination with ancillary soil and meteorology data, to produce 250m resolution soil moisture map over sites in Europe [16]. Torres-Rua et al. (2016) combined Normalized Difference Vegetation Index (NDVI), Leaf Area Index (LAI), energy balance product from Landsat 7, and weather data, and used Relevance Vector Machine (RVM) to relate these potential predictors to SSM [17]. Pause et al. (2012) combined L-band brightness temperature observations and hyperspectral vegetation indices to estimate and improve SSM patterns at the field scale [18]. Qiu et al. (2018) explored the parameterization of SAR vegetation scattering model for high-resolution SSM retrieval with VIs (NDVI, EVI, LAI) and surface roughness derived from Moderate Resolution Imaging Spectroradiometer (MODIS) and Landsat using both the Advanced Integral Equation Model (AIEM) and the Water Cloud Model (WCM) [19]. Klinke et al. (2018) used plant characteristics and temperature as indicators from Sentinel (1, 2) and Landsat archives to derive a high spatial resolution soil moisture product for wetlands in northeastern Germany [20]; same potential of coupling Sentinel 1 and 2 for soil moisture downscaling has also been examined by El Hajj et al. (2017) [21]. Dabrowska-Zielinska et al. (2018) estimated wetland SSM using Sentinel-1 data and addressed the vegetation effect on Radar backscattering change under different SSM and NDVI conditions. In her study, she pointed out that vegetation has a different influence on the backscattering of different polarizations, depending on measurements under dry (soil moisture < 30 vol. %) or moist conditions (soil moisture > 60 vol. %) [22]. Additionally, Samaniego et al. (2010) highlighted the issue of over parameterization and ineffectiveness in integrating spatial heterogeneity in multiscale hydrological models and proposed a multiscale parameter regionalization technique (MPR) to link the dominant process parameter with the finer resolution input data through upscaling operators such as the harmonic mean [23]. Therefore, the work discussed in this paper is based on these critical efforts and aims to address the current SSM monitoring challenges with innovative approaches.

This work mainly focuses on obtaining vegetation information from fine spatial resolution optical EO data and using this information to understand the influence of vegetation on the estimation of spatially varying SSM. The downscaling efforts allow a closer examination of SSM variations over the wetland ecosystem. Specifically, the paper addresses the research question of whether the spatial-temporal heterogeneity in vegetation traits as observed by Sentinel-2 data can be an indicator for SSM as represented by the ASCAT-SWI product. The results illustrate a link between fine-resolution vegetation traits and the soil humidity conditions in wetland environments and demonstrate the potential of using vegetation as sensors for SSM. The work also highlights the commonly used vegetation indices (VIs) and their usability in uncovering the spatial and seasonal relationships between

vegetation and SSM on a regional scale. The paper first describes the current progress and gaps in SSM remote sensing, characterizes the study site selection and experiment workflow, highlights the results, provides a discussion on the results, ecological relevance, and limitation, and then concludes the study for future implications.

## 2. Materials and Methods

### 2.1. Study Areas

The study area is the Okavango Delta, located in northern Botswana between −18.23 and −18.51 °S, 21.84 and 23.81 °E, shown in Figure 1a. The size of the delta is approximately 16,000 km², varying between dry and wet seasons. The climate of the surrounding area is semi-arid; the annual average precipitation ranges from approximately 400 to 500 mm, and the mean annual temperature ranges from 15~20 °C. The wet season usually begins in December, peaks in January and February, and finishes by March. Water infiltrates the Okavango Delta through the Okavango River from the Angolan Plateau in the Northwest. In Figure 1a, the locations of the in-situ water level stations Mohembo (North) and Guma (South) are marked.

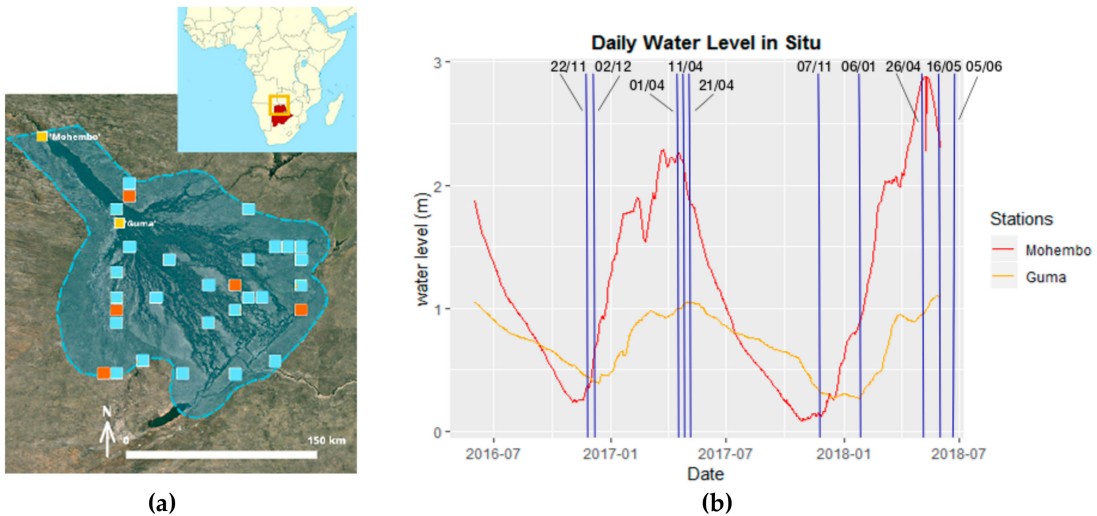

| (a) | (b) |

**Figure 1.** Study area and the regional flow dynamics: (**a**) Extent of the Okavango Delta and the locations of the in-situ water stations (yellow); the locations of five experimental sites (red) and the extended sites (cyan) are indicated; (**b**) Water level records from 2016-06 to 2018-06 measured at Mohembo (red line) and Guma (yellow line) stations, and blue vertical lines indicate the dates of Sentinel-2 imagery and ASCAT-SWI data retrieval.

### 2.2. Data and Pre-Processing

The Sentinel-2A Level-1C products used in this research were obtained from the United States Geological Survey (USGS) EarthExplorer [24]. This cloud-free multispectral imagery corresponds to ten dates (2016-11-22, 2016-12-02, 2017-04-01, 2017-04-11, 2017-04-21, 2017-11-07, 2018-01-06, 2018-04-26, 2018-05-16, 2018-06-05), and they represent the regional dry and wet seasons (Figure 1 b). For each date, six 100 km × 100 km tiles were retrieved for the study area. The Level 1C Top-of-Atmosphere (TOA) reflectance data were first resampled to the spatial resolution of Band 2 (Table 1), and then corrected for atmosphere and cirrus in the Sen2Cor processor (version 2.8.0) distributed by the Sentinel Toolbox Exploitation Platform (STEP) with its Graph Processing Tool (GPT) [25]. The corrected results were reformatted to Level 2A Top-of-Canopy (TOC) reflectance data and subset to spatial extents of the sample sites illustrated in Figure 1a.

**Table 1.** Overview of primary and ancillary datasets used in the research.

| Data | Spatial Resolution | Temporal Coverage | Source |
|---|---|---|---|
| Sentinel-2A Level-1C | 10 m (Band 2,3,4,8) 20 m (Band 5,6,7,8a,11,12) | 11/22/2016~06/05/2018 | Earth Explorer |
| ASCAT-SWI | 25 km | 11/22/2016~06/05/2018 | CGLS |
| PROBA-Vegetation (PROBA-V) Land Cover | 100 m | 2015 | CGLS |
| In-situ Water Level | Site-based | 06/01/2016~06/01/2018 | Okavango Delta Monitoring and Forecasting |

The daily ASCAT-SWI data or simply SWI data, along with the quality flag and metadata, were retrieved from Copernicus Global Land Service (CGLS) as Network Common Data Form (NetCDF) files for the same dates as Sentinel-2 data [9,10]. The spatial resolution of the SWI product matches the SSM by ASCAT [9,10,26]. On board the MetOp satellite series, ASCAT is a real aperture radar instrument, and the scatterometer radar signals can penetrate the surface, thus allowing the detection the subsurface climate feature such as soil wetness [26–28]. The instrument operates during day and night, under all weather conditions, hence the rapid global coverage. The processing of the NetCDF files was done in RStudio.

The in-situ water level records measured at two stations, Mohembo (Latitude: −18.275733, Longitude: 21.787312) and Guma (Latitude: −18.96266, Longitude: 22.373213), were retrieved from the Okavango Delta Monitoring & Forecasting service at daily resolution [29]. The missing data were interpolated with the Kriging method in RStudio. The water level records at both gauges in Figure 1b show clear seasonality. As previously discussed, the Delta region receives a low amount of precipitation; thus, the majority of the water supplied to the Delta is largely related to the Okavango River runoff measured at the two inlet stations. Hence, the wet season in the scope of this research is defined as the time of the year at which high water levels occur at both stations in April through June; dry season is when the water level is low at both gauges in November through January.

The 100m resolution Dynamic Land Cover map of Africa was also obtained from CGLS. The product was derived from the PROBA-V time series for the year 2015 over the continental Africa [30]. The discrete land cover classification and the cover fraction layer for seasonal inland water areas were used in this study to identify the dominant land cover type of each sample site and to eliminate sites with a large water extent that would pollute the SWI signals. The dominant land cover type for each site is the discrete class that has the highest percent coverage at the site.

*2.3. SWI Products and Sample Sites*

SSM can be directly estimated from ASCAT observations at daily temporal resolution, but profile soil moisture cannot be directly measured by remote sensing [26]. To gain insight into the moisture condition beyond the surface soil layer, a relationship between surface and profile soil moisture has to be established, and Wagner et al. (1999) developed a two-layer water balance model to describe this relationship as a function of time [26]. The ASCAT-SWI product was developed within this framework by using the moisture conditions for different characteristic time lengths to represent different depths. Furthermore, the SSM over the preceding time period, *T*, was summed and exponentially weighted. *T* determines how fast the weight becomes smaller and how strongly the SSM observations taken in the past influence the current SWI [26]. The selection of *T = 10 days* was found to be suitable for estimating the influence of recent SSM measures on the SWI [28]. Since the described model was designed to be independent of soil texture and does not involve any vegetation information in its calculation [9,10,26,27], a correlation analysis between the SSM product and the vegetation traits was appropriate to conduct.

A total of 30 sample sites are selected across the Delta (Figure 1). Each site's spatial extent corresponds to a 25km resolution SWI cell and is referenced by the CGLS SWI product cell number [9]. Therefore, five experimental sites were selected in a first step, based on their geographical locations in the Delta and their land cover types to serve as a proof of concept for the workflow developed. Then, 25 additional sample sites were randomly selected across the Delta and evaluated regarding their

suitability to be analyzed with the same methods as the experimental sites. Two sites were excluded manually due to their high cover fraction of inland seasonal water. A total of 28 sites were analyzed in the study for all ten dates with cloud-free images. This workflow, including data preprocessing, SWI retrieval, VI calculation, and correlation analysis, cannot be applied to every SWI pixel because some SWI pixels crossing the Sentinel-2 tiles were cloud-covered or flooded. Therefore, an analysis of continuous spatial coverage is not feasible.

### 2.4. Vegetation Indices (VIs) Retrieval

Four VIs, NDVI, NDWI, LAI, and FAPAR, were retrieved for the six tiles from the pre-processed Sentinel-2 data over the 10 dates (Figure 1b). VIs were batch-calculated with SNAP GPT [25] using the bands listed in Table 2. NDVI measures the photosynthetic activity of vegetation and describes the vitality of vegetation on the Earth's Surface [31–33]. It is included here to correlate with SWI and analyze whether the vitality and greenness of the vegetation are related to the soil water content. The algorithm for calculating NDVI is as below:

$$\text{NDVI} = (\text{NIR} - \text{RED})/(\text{NIR} + \text{RED}), \tag{1}$$

**Table 2.** Sentinel-2A Level-1C spectral bands and center wavelength used for VI retrieval.

| Calculated VIs | Spectral Bands | Central Wavelength (nm) | Band Width (nm) |
|---|---|---|---|
| LAI, FAPAR | B3 Green | 560 | 35 |
| NDVI, LAI, FAPAR | B4 Red | 665 | 30 |
| LAI, FAPAR | B5 Vegetation Red Edge | 705 | 15 |
| LAI, FAPAR | B6 Vegetation Red Edge | 740 | 15 |
| LAI, FAPAR | B7 Vegetation Red Edge | 783 | 20 |
| NDWI, NDVI | B8 NIR | 842 | 115 |
| LAI, FAPAR | B8a Vegetation Red Edge | 865 | 20 |
| NDWI, LAI, FAPAR | B11 SWIR1 | 1610 | 90 |
| LAI, FAPAR | B12 SWIR2 | 2190 | 180 |

NDWI is another important index that measures the liquid water content in the canopy that interacts with the incoming solar radiation [34]. It was included in this study to analyze whether the water content in vegetation is related to the water content in soil. NDWI generally increases as the vegetation fractions and the leaf layer increase, while NDWI is generally negative in areas with naked soil [34]. Gao suggested NDWI contains information independent of NDVI [34]. The equation followed to calculate NDWI is as follow, and Band 11/ SWIR1 is used (Table 2):

$$\text{NDWI} = (\text{NIR} - \text{SWIR}) / (\text{NIR} + \text{SWIR}), \tag{2}$$

LAI and FAPAR are both calculated with the neural networks built in the Biophysical Processor of SNAP software [25,35]. LAI is defined as half the developed area of photosynthetically active elements of the vegetation per unit horizontal ground area. It is used to determine the size of the interface for energy and mass exchange between canopy and atmosphere [36,37]. FAPAR measures the fraction of photosynthetically active radiation absorbed by the canopy, and it corresponds to the canopy's primary productivity of photosynthesis [37,38]. Both VIs were analyzed to understand the vegetation's evapotranspiration and photosynthetic primary production capacity as related to SSM. Based on SNAP algorithm descriptions, to calculate each input biophysical variable (LAI or FAPAR), the neural network is trained with a representative set of TOC reflectance and with prior information on the distribution of the input variables from the training data. After adjusting the synaptic weights and neuron bias according to a combination of tangent sigmoid and linear transfer functions, the trained neural network can then be used in operational mode for new calculation. The network takes 11 normalized input data including 8 Sentinel-2 TOC reflectance wavebands (B3, B4, B5, B6, B7, B8a, B11 and B12) and the

geometry of acquisitions ($\cos(\theta_s)$, $\cos(\theta_v)$, and $\cos(\theta_\varnothing)$) to output targeted biophysical variable (LAI or FAPAR) for each pixel [35].

*2.5. Statistics Retrieval*

The mean value, standard deviation (SD), and coefficient of variation (CV) are calculated as statistical parameters of the VIs over each study site and each date of analysis (Figure 1) to capture the central tendency and spatial heterogeneity of the VIs. Second-order image entropy and homogeneity are used to describe the image texture and to reflect vegetation structure.

Mean ($\mu$) of the VIs is calculated using the following formula:

$$\mu = \frac{\sum x}{n} \, , \tag{3}$$

In (3) $x$ denotes the VI value at each pixel; $n$ indicates the total number of pixels in a given image. SD ($\sigma$) describes the variation of data values in the VIs about the mean using the formula below:

$$\sigma = \sqrt{\frac{\sum |x - \mu|^2}{n}} \, , \tag{4}$$

In (4), $x$ denotes VI value at each pixel; $n$ indicates the total number of pixels in a given image.

CV is the ratio of SD to mean, and it measures the relative variability in the dataset. It adjusts the variation for the mean so it allows comparison across data values from different datasets; in comparison with SD, it highlights the variability in data overshadowed by low SD (5).

$$CV = \frac{\sigma}{\mu} \, , \tag{5}$$

In addition to the standard statistics, the second-order image texture, entropy and heterogeneity, are calculated using the Grey Level Co-occurrence Matrices (GLCM). The image texture of VIs can capture gradients in vegetation structure that may be overshadowed by the discrete land cover [39,40]. GLCM are a statistical texture analysis method that describe the spatial distribution of the observed intensity pairs in respect to their relative distances [40]. Entropy and homogeneity are selected among many statistical measures derivable from GLCM to represent the orderliness and contrast group within the second-order image texture measures. Entropy mainly measures the disorderedness of the image pixel and when GLCM has the same values, the entropy is the highest (6). Homogeneity measures the closeness of the distribution of elements to the GLCM diagonal (7). The mean and SD for entropy and homogeneity are calculated to summarize the texture analysis for each VI using the "glcm" package in R. The calculations use N = 32 as the number of grey levels for all directions (0 degrees, 45 degrees, 90 degrees, and 135 degrees), and a 3 × 3 window size. This window size has the advantage of capturing the heterogeneity of pixel values over a small distance [39,40].

$$entropy = \sum_{i,j=0}^{N-1} -ln\left(P_{ij}\right)P_{ij} \, , \tag{6}$$

$$homogenieity = \sum_{i,j=0}^{N-1} \frac{P_{ij}}{1 + (i-j)^2} \, , \tag{7}$$

where $i$ is the row number and $j$ is the column number. $P_{ij}$ is the probability value recorded for the cell $i$, $j$; N is the number of rows or columns.

## 3. Results

### *3.1. SWI and VIs Correlation by Season*

In terms of the means of most VIs, stronger positive correlations are noted for wet season observations (Figure 2). Correlations for wet seasons are slightly higher than those for mixed observations from both seasons but generally, they are not very distinct. However, it is obvious to see the dry season observations show little positive linear correlation to SWI. In the case of LAI, a slightly negative correlation can be observed but the correlation is not significant. This indicates that mean values of four VIs are positively correlated to SWI in the wet season comparing to the dry season, meaning dry season vegetation conditions vary greatly over these sample sites.

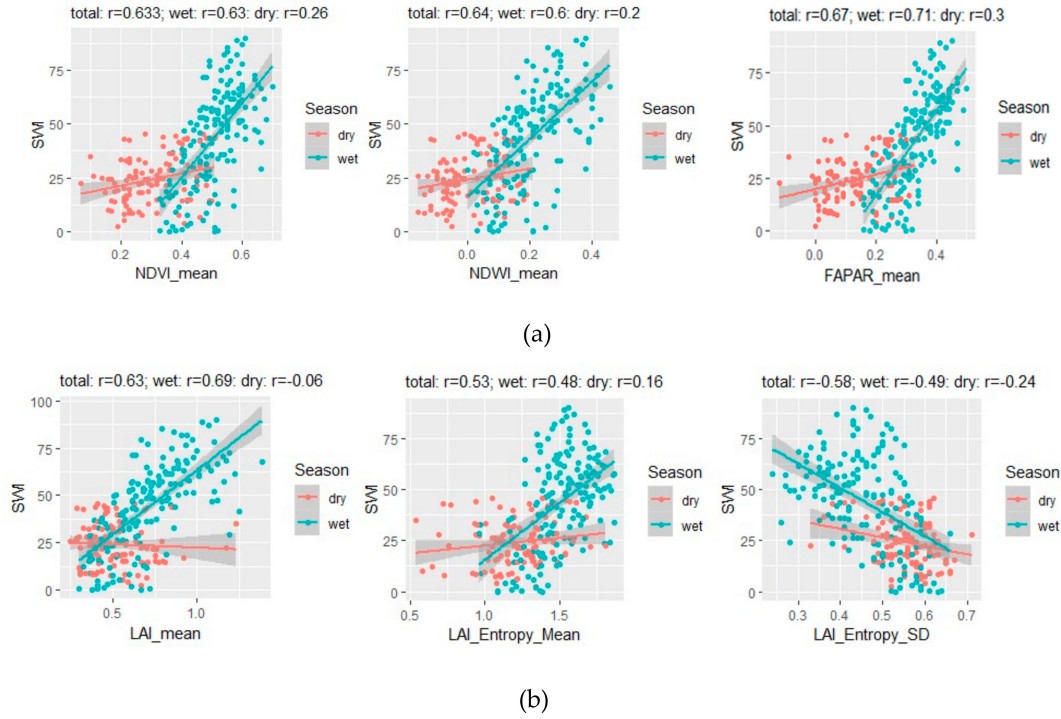

**Figure 2.** Selected scatterplots showing the distribution of observation by season, and the relationship between SWI and key VI statistics for the 28 samples sites across the Delta. Only the top-performing statistics (in terms of r values) are shown here. Blue indicates wet season and red for dry season. Grey lines display the confidence interval at 0.95. In each panel: (**a**) Correlating NDVI, NDWI, and FAPAR mean to SWI; (**b**) Correlating LAI mean, LAI entropy mean and SD to SWI.

In the SD plots for the four VIs, a moderately positive correlation (> 0.50) is observed for the wet seasons and weak negative correlation is observed for the dry seasons (Figure 2). This indicates a difference between the variations of vegetation conditions in different seasons and how the variation correlates to soil humidity conditions. Moreover, this demonstrates, as expected, that higher variation in activated vegetation cover during the wet season occurs in association with increased variance in soil moisture conditions as captured through the ASCAT-SWI. This is also verified by the observation that the higher the variation in vegetation's leaf surface area or the area of photosynthetic activity becomes, the higher the SWI.

CVs depicting relative variability indicate a stronger negative linear correlation, r = −0.4 for NDVI SD and r = −0.21 for LAI SD. This demonstrates a higher absolute deviation of the individual values within a cell in a high moisture saturation period, but a higher relative variability in NDVI and LAI in a lower moisture saturation period. Because NDWI and FAPAR can get 0 or negative values, a CV calculation for NDWI and FAPAR cannot be performed.

Texture information (entropy and homogeneity) also contributes to the explanation of the variance in SWI/VI relations but to a smaller extent (Figure 2, Table 3). The mean and SD values for entropy (calculated from the GLCM) indicate the level of disorder in the VI distribution, and smooth image values result in high entropy. The mean entropy of LAI shows a positive linear correlation with SWI at r = 0.53; the SDs of entropy for LAI are negatively correlated to SWI at r = −0.58. The FAPAR entropy SD (r = −0.48) for the dry season also shows a moderately negative correlation with the SWI (r = −0.48). Therefore, moderate-to-weak correlations can be found between soil moisture condition and the second-order texture measures of variables estimating vegetation's evapotranspiration and photosynthetic primary production capacity.

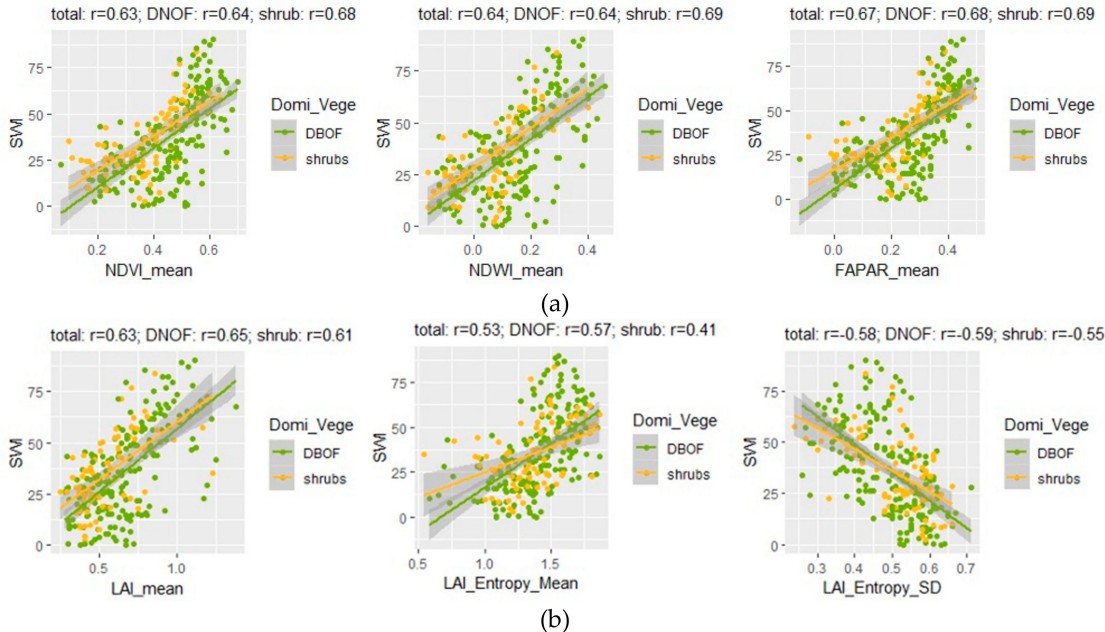

**Figure 3.** Selected scatterplots showing the distribution of observations by dominant land cover types, and the relationship between SWI and key VI statistics for the 28 samples sites across the Delta. Only the top-performing statistics (in terms of r values) are shown here. Green indicates observations with DBOF as dominant vegetation and yellows are for shrubs. The grey lines display the confidence interval at 0.95. In each panel: (**a**) Correlating NDVI, NDWI, and FAPAR mean to SWI; (**b**) Correlating LAI mean, LAI entropy mean and SD to SWI.

**Table 3.** Remaining top performing statistics (in terms of r values) describe the second-order homogeneity (mean and SD) that are not illustrated in Figures 2 and 3.

| VI_Stats | Total r | Total p | Wet r | Wet p | Dry r | Dry p | DBOF r | DBOF p | Shrub r | Shrub p |
|---|---|---|---|---|---|---|---|---|---|---|
| NDVI_hom_SD | −0.39 | $1.16 \times 10^{11}$ | −0.30 | $9.39 \times 10^5$ | −0.35 | 0.000148 | −0.36 | $1.09 \times 10^7$ | −0.49 | $3.97 \times 10^6$ |
| LAI_hom_mean | −0.50 | $2.43 \times 10^{19}$ | −0.42 | $1.57 \times 10^8$ | −0.17 | 0.075967 | −0.55 | $5.20 \times 10^{17}$ | −0.38 | 0.000497 |
| FAPAR_hom_SD | −0.38 | $3.49 \times 10^{11}$ | −0.19 | 0.015037 | −0.35 | 0.000176 | −0.36 | $2.11 \times 10^7$ | −0.47 | $9.49 \times 10^6$ |

The observations for the two seasons do not form completely separate clusters. The lower range of the wet season observations intermingles with part of dry season at mid to low signals. This may be related to the contribution of other water sources, such as groundwater and precipitation, in vegetation conditions, but they are not in the scope of the wet/dry season definition in this research.

*3.2. SWI and VIs Correlation by Dominant Land Cover Type*

Two land cover types, deciduous broad open forest (DBOF) and shrubs, are dominating the sites based on the CGLS 2015 Africa Land Cover product [30]. Overall, the distribution of observations for

both classes does not result in separate clusters (Figure 3). This indicates that these two dominant land cover types behave similarly with respect to soil moisture dynamics.

Mean plots for all four VIs show a strong positive linear correlation with the SWI (r > 0.60). The shrub-dominant sites indicate a strong correlation with SWI (r > 0.68) in terms of mean NDVI, NDWI, and FAPAR; DBOF-dominant sites also show a strong correlation. Spatial variance described by SD and CV are moderate in correlation strength with SWI for NDVI, NDWI, and FAPAR. The SD of FAPAR shows weak to no correlation with SWI. Second-order entropy in Figure 3 and second-order homogeneity in Table 3 both show weak-to-no correlation except for LAI, where a moderate correlation strength can be observed. This indicates a partial significance of second-order image texture measures in the discussion of vegetation structures relevant indices, such as LAI, over the study area.

## 4. Discussion

The series of correlation analyses demonstrates the possibility of using VIs to downscale SSM in the wetland environment. Seasonal differences in using vegetation proxies for soil moisture are obvious—in the wet season, vegetation information has a strong linkage to soil moisture condition, while very scattered results are obtained during the dry season. Different reactions of dominant vegetation types in respect to the soil moisture distributions exist, but they are not drastic in the Okavango study sites—at sites with shrubs as dominant vegetation, vegetation proxies performed generally well in estimating soil moisture; at sites with DBOF as dominant vegetation, a moderately strong correlation could be found as well.

The mean values for all VIs correlate moderately to strongly with SWI. The FAPAR mean is marginally stronger than the rest. An interpretation is that the vegetation's evapotranspiration and photosynthetic primary production capacity is well linked to SSM. LAI and FAPAR are closely linked biophysical variables that characterize the total canopy and the photosynthetic activity of plants., while LAI, only accounting for the amount of foliage in the plant canopy including the understory, FAPAR, reveals more of the amount of light absorbed by canopy at a given time. Based on the crop-specific empirical relationship between these two indices analyzed by Kukal and Irmak (2020), the increasing in the leaf area is accompanied with the linear increase of light absorption by the plant but the linearity diminishes at a threshold (LAI of 2-4) and this diminishing return denotes that FAPAR is a more direct proxy of vegetation's light absorption capacity than the canopy area [41]. Moreover, the vegetation's vitality and greenness, conveyed through the NDVI, and NDWI, which measures the liquid water content in vegetation, both correlate moderately to strongly with SWI.

SD and CV measure first-order spatial variation in the four VIs. They are moderately correlated with SWI (Figure 2, Figure 3). CVs are generally negatively correlated to SWI while SDs are positively correlated to SWI. This indicates that spatial variability in the VIs could be meaningful in understanding VIs' relations with SWI. The texture information, homogeneity, and entropy, correlated weakly to moderately to SWI. These second-order texture measures describe properties of the horizontal vegetation structure through the relationship comparison of directly neighboring pixels. They are mathematically more complex than the first-order statistics discussed above but can reveal particular patterns. In the wet season where foliage is abundant, textural information is more distinct since the original signals are more dynamic. Correlation strength with SWI is, therefore, stronger in comparison to the dry season. In comparing the four selected VIs, textural information for LAI is consistently more relevant in its correlation with SWI.

The stratification strategy both by season and by dominant land cover type in the study of the relationship between VIs and SWI encompasses a temporal and a spatial aspect. The observation dates were selected firstly to reflect two states of the regional flow dynamics via measured in-situ water levels and, secondly, to reflect the availability of cloud-free Sentinel-2 images for the target seasons. However, analyses of the temporal behavior are limited to a comparison of peak season variation by grouping these dates into dry and wet season data, because an evenly spaced time series could not be constructed. Regarding the dominant land cover selection, among the 28 sample sites, 20 sites have DBOF as the

dominant type and eight sites have shrubs. When two competing land cover types appear within one site, the one with the higher coverage has been defined as the dominant type. The signals observed in VIs, however, do not purely stem from one uniform type of vegetation. In a fairly heterogeneous landscape as the Okavango Delta, it is unlikely to find one site with a uniform vegetation type to be used as a stereotype. Such purity would have revealed a closer correlation between vegetation status and SWI. Nevertheless, some distinction was possible in defining dominant vegetation.

The results indicated in this study align with the conclusion of Torres-Rua et al. (2016) that one single vegetation index, such as NDVI, is insufficient to describe the internal variance of SSM fully; the addition of other vegetation indices (such as LAI) and spatial heterogeneity parameters can provide an improved spatial estimation of SSM [17]. In this study, different characteristics of each investigated VI infer that the heterogeneity in the vitality and evapotranspiration of vegetation, and the photosynthetic primary production capacity over the landscape contribute to the explanation of the SSM patterns underneath. As Figure 2; Figure 3 indicate, the mean values of vegetation characteristics perform best among all statistical descriptors of vegetation conditions in correlating the most strongly with SWI. This result is consistent with Klinke et al.'s (2018) findings [20] and is responsive to the multiscale parameter regionalization (MPR) technique proposed by Samaniego et al. (2010) to downscale coarse resolution parameters with finer resolution input data through upscaling operators like the harmonic mean [23]. Additionally, Klinke et al. (2018) pointed out the long-range and intra-annual variations in SSM [20], and Dabrowska-Zielinska et al. (2018) analyzed the different contributions of vegetation in dry and wet moist conditions of soil on backscattering [22]. Both indicate the seasonal effect in vegetation–soil relations. This idea has been taken up through a dry/wet season observation stratification in this study, whereas a stronger linear correlation between VI and SWI can be found for the wet season observations. However, no obvious differences could be found with respect to dominant vegetation types based on the 2015 PROBA-V Land Cover map of Africa [30]. This may be due to similarities in the reaction of these two dominant savanna vegetation classes. Moreover, this demonstrates that real-time vegetation information demonstrates a stronger capability to estimate SSM while non-real-time vegetation information from past land cover map does not add much information to the estimation of SWI in this study. To further understand how the near real-time vegetation traits are associated with soil moisture condition, long term time series of VIs and time-lagged analysis should be used to interpret when the vegetation is stressed after some accumulation in a plant's water-limited features over time.

## 5. Conclusions

The remote sensing analyses in this study make use of the popular remote sensing products, ASCAT-SWI for SSM and the Sentinel-2 images for VI retrieval. They demonstrate the relationships between vegetation and soil moisture in a complex wetland environment and the potential of using vegetation proxies from multispectral data for downscaling and regionalization of soil moisture. Fine spatial resolution vegetation traits calculated from the Sentinel-2 data convey information about the internal moisture variation within each coarse SWI pixel and indicate that both the absolute VI values and the individual spatial variation in the vegetation structure relate to soil moisture conditions at the time of VI retrieval. The stratification techniques, by seasons and by dominant land cover type, also shed light on how the soil-vegetation relations change under the influence of regional flow dynamics and dominant vegetation patterns. Finally, it can be inferred that the time when vegetation flourishes are clearly reflected through high values in VI signals and are associated with stronger correlations with soil humidity; low VIs, on the other hand, indicating low vegetation vitality, have a very limited correlation with SWI. In sites with different dominant vegetation, correlations strength does not differ much, but vegetation structures could make an influential distinction. The latter needs further analysis. Future scientific efforts are needed to understand vegetation signals' delayed response to changes in soil moisture; this is hindered in this research by a limited number of cloud-free Sentinel-2 data. Long-term and high temporal resolution time series of vegetation traits, however,

will have great potential in uncovering the relation between coarse-scale soil moisture and fine-scale vegetation. Therefore, the research exhibits elements in line with state-of-the-art soil monitoring in remote sensing and informs that vegetation data should be implemented in soil moisture retrieval algorithms to improve the estimation of spatially varying soil moisture. Further space missions collecting hyperspectral data and terrestrial vegetation's chlorophyll fluorescence data from the FLuorescence EXplorer (FLEX) can also be integrated into this line of studies to provide more detailed information on the linkage of vegetation traits and soil water availability.

**Author Contributions:** Conceptualization, M.L. and M.P.; Formal analysis, M.L.; Investigation, M.L.; Methodology, M.L.; Software, M.L.; Supervision, M.P., N.P. and M.S.; Visualization, M.L.; Writing—original draft, M.L.; Writing—review & editing, M.P., N.P. and M.S. All authors have read and agreed to the published version of the manuscript.

**Acknowledgments:** Thanks are extended to the technical support this work received from the Institute of Photogrammetry and Remote Sensing and Institute of Cartography at TU Dresden, as well as the support in editing the written work received the University of Maryland Graduate School Writing Center.

**Conflicts of Interest:** All authors declare no conflict of interest.

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
