# Peer review of "Regionalization of Coarse Scale Soil Moisture Products Using Fine-Scale Vegetation Indices—Prospects and Case Study"

_remotesensing, doi:10.3390/rs12030551_

Round 1

Reviewer 1 Report

Note about remotesensing-696978
The paper presents a method for using vegetation indices from earth observation data for an indirect characterization of surface soil moisture conditions. This study demonstrates that there is a correlation between vegetation indices and soil water index, independently from the underlying dominant vegetation. Information from optical satellite data can convey the spatial heterogeneity missed by coarse surface soil moisture products.
The goal of the study has a relevant scientific impact and various benefits in the perspective of the potential use of vegetation proxies from multispectral data for downscaling of soil moisture.
I recommend the publication of this interesting manuscript after the following minor revisions.
*****************************************
General comments
• Sometimes the acronyms are not explained (e.g. SAR): I suggest explaining them all and capitalize the first letters of the words of acronyms for clarity.
• Some acronyms are explained only in the abstract. It would be better if they were also explained the first time they are mentioned in the text.
• ASCAT-SWI and SWI are the same? If so, I suggest making the term uniform in all the text.
*****************************************
Specific comments
1. Introduction
• The authors mention large scale and point scale soil moisture monitoring: I suggest mentioning also techniques that fill the gap between the two like proximal gamma-ray and cosmic-ray neutron methods. These works can be referenced:

o Zreda, Desilets, D., Ferré, T., Scott, R.L., 2008. Measuring soil moisture content non- invasively at intermediate spatial scale using cosmic-ray neutrons. Geophys. Res. Lett. 35 (21). http://dx.doi.org/10.1029/2008GL035655
o Filippucci, P., A. Tarpanelli, C. Massari, A. Serafini, V. Strati, M. Alberi, K. G. C. Raptis, F. Mantovani & L. Brocca (2020) Soil moisture as a potential variable for tracking and quantifying irrigation: a case study with proximal gamma-ray spectroscopy data. Advances in Water Resources, 103502. http://dx.doi.org/10.1016/j.advwatres.2019.103502
2. Materials and Methods
• I suggest explaining the provenance of the data: it says that they are a remote sensing product but it is never mentioned in the text (only in the abstract) that are satellite data.
• Lines 147-152: the explanation of how SWI is calculated and what it is it’s not entirely clear. Also, the difference between SWI and SSM is vague. Please change these lines to explain better.
• Line 168: “Retrivel” should be “Retrieval”.
• Line 179: “soil” should be “naked soil”.
• Line 202: remove “below”.
• Please describe the variables in the formulas (e.g. in formula 3 what is x? And n?).
• Lines 207-223: there is a maybe too technical description of second-order image texture, entropy and
heterogeneity. I suggest an introduction for a less technical reader who is just approaching this topic.
3. Results
• Line 233-234: the authors cite “SD plots” and “LAI SD” in fig.2 but in that figure I can only find LAI
entropy SD.
• Figure 2: the FAPAR mean graph is shorter than the others.
• Line 271: I suggest removing “while”.
4. Discussion
• Line 348: does real-time vegetation information demonstrate or just imply a stronger capability to
estimate SSM?

Author Response

Point 1: General comments
• Sometimes the acronyms are not explained (e.g. SAR): I suggest explaining them all and capitalize the first letters of the words of acronyms for clarity.
• Some acronyms are explained only in the abstract. It would be better if they were also explained the first time they are mentioned in the text.
• ASCAT-SWI and SWI are the same? If so, I suggest making the term uniform in all the text.

The acronyms are now spelled out as indicated.

The ASCAT-SWI and SWI are the same; the SWI is ASCAT-SWI shortened for concision. This relationship is now clarified in the abstract and in the dataset description.  

Point 2: 1. Introduction
• The authors mention large scale and point scale soil moisture monitoring: I suggest mentioning also techniques that fill the gap between the two like proximal gamma-ray and cosmic-ray neutron methods. 

The suggested literature have been reviewed, and the techniques described for intermediate spatial scale soil moisture monitoring have been mentioned in the introduction section.

Point 3: Materials and Methods
• I suggest explaining the provenance of the data: it says that they are a remote sensing product but it is never mentioned in the text (only in the abstract) that are satellite data.

The fact that the datasets used in this research are from satellite remote sensing is now explicitly stated. 

Point 4:• Lines 147-152: the explanation of how SWI is calculated and what it is it’s not entirely clear. Also, the difference between SWI and SSM is vague. Please change these lines to explain better.

This paragraph explaining the estimation of SWI from SSM measurements are now reworded to increase clarity. 

Point 5:• Line 168: “Retrivel” should be “Retrieval”.

Corrected.

Point 6:• Line 179: “soil” should be “naked soil”.

Corrected.

Point 7:• Line 202: remove “below”.

Corrected.

Point 8:• Please describe the variables in the formulas (e.g. in formula 3 what is x? And n?).

The variables in the formula 4 and 5 are now described. 

Point 9:• Lines 207-223: there is a maybe too technical description of second-order image texture, entropy and
heterogeneity. I suggest an introduction for a less technical reader who is just approaching this topic.

According to the cited literature on the gray level co-occurrence matrix (GLCM), the included text is necessary to describe the basis of the concept. The technical parameters for implementing the GLCM are included for reproducibility. Thus, the section remains as it is, but the author appreciates this comment.

Point 10:3. Results
• Line 233-234: the authors cite “SD plots” and “LAI SD” in fig.2 but in that figure I can only find LAI
entropy SD.

The text is now adjusted to match the graph annotations in fig.2.

Point 11:• Figure 2: the FAPAR mean graph is shorter than the others.

The graph is resized to match other graphs.

Point 12:• Line 271: I suggest removing “while”.

Corrected.

Point 13: Discussion
• Line 348: does real-time vegetation information demonstrate or just imply a stronger capability to
estimate SSM?

Real-time vegetation information demonstrates a stronger capability to
estimate SSM; The wording has been modified.

Reviewer 2 Report

The introduction provides clearly all the critical elements of an ideal introduction, including background, problem, knowledge gap, other contemporary efforts, and specific objectives.

In figure 1: are the numbers within the squares meant to be mean something. If yes, please try to increase the visibility of the figure. Currently, these are illegible. Or the numbers can be removed, since you already define the colors in the caption. Adding a legend to the figure itself might be useful too, but I’ll leave this decision to the authors.

Also, please define the blue vertical lines in figure 1 b. The caption would be a good place to do so.

Line 295-303: The differences among LAI and FAPAR at the ground scale are substantial. While LAI only represents the leaf area or foliage, the FAPAR, in fact represents the actual amount of light absorbed by the canopy at any time (relative to incident light). The relationship among LAI and FAPAR is a diminishing return curve. More LAI in a canopy is not proportionally (linearly) related to the FAPAR in the canopy. Kukal and Irmak (2020) listed below has a curve that you can refer to, and possibly follow them to construct your own FAPAR vs. LAI curve from your satellite data. Thus, FAPAR is more reflective of the actual light absorbing nature of canopy than LAI. This is more likely a reason for your observation of FAPAR better reflecting SWI dynamics.

They both have little to do directly with photosynthesis, which is a function of other inherent vegetation properties like stomatal conductance. Thus, the statements made in this paragraph are oversimplification of canopy dynamics.

Please account for the abovementioned reasons in this discussion paragraph.

Here is the study that you can follow:

Kukal, M. S., & Irmak, S. (2020). Light interactions, use and efficiency in row crop canopies under optimal growth conditions. Agricultural and Forest Meteorology284, 107887.

Author Response

Point 1: In figure 1: are the numbers within the squares meant to be mean something. If yes, please try to increase the visibility of the figure. Currently, these are illegible. Or the numbers can be removed, since you already define the colors in the caption. Adding a legend to the figure itself might be useful too, but I’ll leave this decision to the authors.

Also, please define the blue vertical lines in figure 1 b. The caption would be a good place to do so.

The map indicating the study site locations have been modified to remove the numbering. The blue vertical lines are also described in the caption below.

Point 2: Line 295-303: The differences among LAI and FAPAR at the ground scale are substantial. While LAI only represents the leaf area or foliage, the FAPAR, in fact represents the actual amount of light absorbed by the canopy at any time (relative to incident light). The relationship among LAI and FAPAR is a diminishing return curve. More LAI in a canopy is not proportionally (linearly) related to the FAPAR in the canopy. Kukal and Irmak (2020) listed below has a curve that you can refer to, and possibly follow them to construct your own FAPAR vs. LAI curve from your satellite data. Thus, FAPAR is more reflective of the actual light absorbing nature of canopy than LAI. This is more likely a reason for your observation of FAPAR better reflecting SWI dynamics.

They both have little to do directly with photosynthesis, which is a function of other inherent vegetation properties like stomatal conductance. Thus, the statements made in this paragraph are oversimplification of canopy dynamics.

The recommended literature has been reviewed, and the LAI-FAPAR relationship discussed in the literature has been added in the discussion of this research results to provide more detailed explanation of the difference between FAPAR and LAI in interpreting Soil Moisture.